# Assessment of Five Questionnaires for Chronic Obstructive Pulmonary Disease in a Southern Italian Population: A Proof-of-Concept Study

**DOI:** 10.3390/medicina59071252

**Published:** 2023-07-05

**Authors:** Silvano Dragonieri, Sean Galloway, Vitaliano Nicola Quaranta, Andrea Portacci, Maria Rosaria Vulpi, Carla Santomasi, Agnese Caringella, Giovanna Elisiana Carpagnano

**Affiliations:** Respiratory Diseases, University of Bari “Aldo Moro”, 70121 Bari, Italy; sean.galloway13@gmail.com (S.G.); vitalianonicola.40@gmail.com (V.N.Q.); a.portacci01@gmail.com (A.P.); mariarosaria86@alice.it (M.R.V.); carlasantomasi@gmail.com (C.S.); agnese.cari@gmail.com (A.C.); elisiana.carpagnano@uniba.it (G.E.C.)

**Keywords:** chronic obstructive pulmonary disease, screening, questionnaire, accuracy

## Abstract

*Background and Objectives*: Chronic obstructive pulmonary disease (COPD) is a growing burden to society, and remains underdiagnosed in Italy. This study aimed at evaluating five validated screening questionnaires to consider which one was the most accurate, and the optimal cut-off score for each to be considered for the Southern Italian population. *Materials and Methods*: A total of 144 patients were recruited in the study. The age range was 46–85 years. All subjects underwent spirometry, and completed the five questionnaires: CDQ, LFQ, COPD-PS, COPD-SQ, and CAPTURE. Receiver-operator curves (ROC) were drawn for each questionnaire. The area under the curve (AUC), sensitivity, specificity, positive predictive value (PPV), negative predictive value (NPV), values for the optimal cut-off score and previously recommended score were calculated and compared. *Results*: Of the questionnaires, the CDQ, LFQ, and COPD-SQ had significant differences between COPD (*n* = 86) and non-COPD (*n* = 52) groups. The AUCs for each questionnaire with (95%CI) were: CAPTURE, 0.602 (0.431–0.773); CDQ, 0.714 (0.555–0.872); LFQ, 0.331 (0.183–0.479; COPD-PS, 0.652 (0.497–0.807); and COPD-SQ, 0.679 (0.520–0.837). Only the CDQ and COPD-SQ had significant AUC screening characteristics. The optimal cut-off values for the CDQ, LFQ, and COPD-PS were modified to 22, 10, and 4, respectively. The COPD-SQ remained at 17. *Conclusion*: The CDQ and COPD-SQ can discriminate between individuals with and without COPD in the Italian population. The CDQ has a moderate screening accuracy, and the COPD-PS and COPD-SQ have low accuracy, when the optimal cut-off scores are used. Of the five questionnaires assessed, the CDQ and COPD-SQ questionnaires could be used for screening for COPD in the Southern Italian population.

## 1. Introduction

Chronic obstructive pulmonary disease (COPD) is defined as a heterogeneous lung condition characterized by chronic and progressive dyspnea, a chronic cough, sputum production, chest tightness, fatigue, dyspnea, and exacerbations [1]. The condition is due to airway and/or alveoli abnormalities which lead to persistent, often progressive, airflow obstructions [1]. The diagnosis of COPD is a clinical judgment made by healthcare professionals based on a combination of factors, including patient history, symptoms, physical examination, and diagnostic test results, such as lung function, blood tests, chest X-ray and high-resolution computed tomography. Interpretation of test results should always be performed in the context of the individual patient’s clinical presentation [1]. COPD continues to be a growing global burden, affecting around 10.3% of the world’s population [2], with current estimates stating that over 50% of the patients are undiagnosed [2]. COPD is also projected to become the third leading cause of mortality by the year 2030, and the seventh-highest in terms of burden of disease [3,4] Within the Italian population, the Superior Health Institute states that respiratory illness was responsible for 6.4% of total deaths in 2002, 50% of which are attributed to obstructive diseases [5]. One cross sectional epidemiological study undertaken in northern Italy found that 6% of men aged 46–55, and 11% of men aged 55–65, had been diagnosed with chronic bronchitis and, using smoking rates, it was suggested that some 2.6 million men and women aged 45–70 years of age have COPD [6]. The overall trend of the disease burden is also set to increase in the coming years, with advances in population demographics in Italy having the third-oldest population worldwide, coupled with an average life expectancy of 79.7 (males) and 84.4 (females) years [7], and with improvements in the diagnosis and treatment of other causes of death, such as cardiovascular disease, leading to potentially over 5.4 million deaths due to COPD and related diseases worldwide by 2060 [1]. The high impact of COPD on the health system in Italy is a combination of both underdiagnosis and, therefore, undertreatment of the disease. The reasons for the above can be due to diagnostic challenges, since access to spirometry tests may be limited in some healthcare settings, not all primary care physicians are trained to interpret the results accurately, and/or to feelings of guilt, shame, fear of being judged by healthcare providers, or to the fragmented Italian healthcare system, which often lacks coordination and integration of care across different healthcare settings. This can result in delayed diagnosis and treatment. This suggests that the financial cost burden may, at least in part, be reduced by finding credible, economically viable health strategies which can be implemented to reduce the principle means by which COPD burdens the health system, such as the need for long term oxygen therapy, high hospitalization rates, and the high proportion of costs related to acquired severe disabilities [8]. The Global Initiative for Chronic Obstructive Lung Disease (GOLD) guidelines state that lung function assessment is the gold standard for diagnosing COPD [1]. However, it is arduous to perform pulmonary function tests for a large number of patients, especially at primary care levels. Therefore, the low spirometry utilization rate may lead to underdiagnosis and/or misdiagnosis of COPD. Population self-administered screening questionnaires could support identifying patients at risk for COPD who need lung function testing [9]. The most frequently used screening questionnaires for COPD include the COPD Diagnostic Questionnaire (CDQ), COPD Population Screener (COPD-PS), COPD Screening Questionnaire (COPD-SQ), COPD Assessment in Primary Care to Identify Undiagnosed Respiratory Disease and Exacerbation Risk (CAPTURE), and the Lung Function Questionnaire (LFQ). The implementation of screening tests for COPD offers numerous advantages, including early detection, improved patient outcomes, targeted risk factor modification, cost-effectiveness, increased awareness and education, and valuable data for research and public health planning. By identifying COPD in its early stages, screening tests can facilitate timely interventions, leading to improved patient outcomes, reduced healthcare costs, and a better overall management of the disease. To date, no previous investigation has assessed the effectiveness of these five questionnaires in screening for COPD in the Italian population, including which of the above questionnaires might be the most suitable for this people. Thus, the aim of the current study was to evaluate the screening accuracy of the five questionnaires in a Southern Italian population, to consider which (if any) of the questionnaires was the most accurate, and the optimal cut-off score for each to be considered for the Southern Italian population.

## 2. Materials and Methods

### 2.1. Patients and Study Design

We conducted a case–control study according to the Declaration of Helsinki guidelines. The study was approved by the local Ethical Committee (protocol number: 5785). We enrolled 144 patients attending to the outpatient clinic of the Respiratory Diseases Institute, University of Bari, Italy, between January 2022 and December 2022. The age range was 46–85 years. All patients satisfied the selection criteria and provided informed consent. Inclusion criteria were the following: age between 40 and 85 years; diagnosis of COPD from less than 2 years and more than 1 year ago, according to GOLD guidelines (history of current or prior smoking, chronic symptoms of sputum production or dyspnea during efforts, post-bronchodilator FEV1/FVC ratio < 70% and absence of clinical asthma or other respiratory and/or cardiovascular abnormalities) [1]; COPD in the stable phase of the disease (previous 8 weeks without exacerbations); an absence of other chronic respiratory diseases; and ability to fill out questionnaires. Exclusion criteria were as follows: refusal to provide written informed consent; previous established diagnosis of COPD; exacerbation in the previous 8 weeks; history of other respiratory diseases and severe cardiac or renal comorbidities; inability to complete questionnaires; inability or unwillingness to complete lung function assessment; inhalation of bronchodilators or corticosteroids on the day before the test. The questionnaires were face-to-face administered during a one-day visit by a trained operator. The order of tests was randomized by self-designed sequence generator software. After completing the set of five questionnaires, all participants underwent lung function evaluation (Figure 1).

### 2.2. Screening Questionnaires

The Lung Function Questionnaire (LFQ) is a five-item questionnaire which, using a cut-off of 18 or below, has a sensitivity of approximately 82% and a specificity of 47% in a primary care setting of current and former smokers [10]. Although, in the validation study conducted by Hanania et al. [10], a higher AUC was higher for a cut-off of 16, it was deemed more beneficial to have the cut-off set at 18 to try to combat the underdiagnosis of COPD, thus avoiding missed diagnosis and allowing disease progression before intervention occurs. The questionnaire itself comprises of five items relating to mucus production, frequency of chest wheeze/whistling/rattles, exertional dyspnoea, number of years smoked and age. Patients are also encouraged to speak to their doctor following the completion of the questionnaire regarding symptoms regardless of their score, providing a direct opportunity to improve patient–physician rapport [10].

The COPD Population Screener (COPD-PS) is a five-item questionnaire; using a cut-off of 5 or greater, it has a sensitivity of 67% and specificity of 73% in a general population in Japan. It was created by Martinez et al. in 2008 [11], with the idea of identifying patients with possible COPD in the general population. According to the authors, the cut-off score may be modified to satisfy particular objectives; that is, a lower threshold score would yield a higher number of patients deemed to have airway obstruction, thus identifying as many potential COPD patients as possible, but reducing the specificity by including patients who do not have an obstruction adequate to merit GOLD classification. On the contrary, the score may be increased if the aim is to ensure, with a high specificity, all patients who meet the threshold indeed have COPD [11].

The COPD diagnostic questionnaire (CDQ) is an eight-item tool designed by the COPD Questionnaire Study Group from a cross-section of patients ≥40 years of age from the UK and USA, with a history of smoking but without a definitive respiratory diagnosis. It was not initially designed as a diagnostic tool and, indeed, does not seem to perform well as such. Following several validation studies, it became apparent, however, that the CDQ has the potential to be utilized as a tool to identify high-risk patients who should undergo further spirometric analysis, and has been recommended by the United States Preventive Services Task Force (USPSTF) [12].

The COPD Assessment in Primary Care to Identify Undiagnosed Respiratory Disease and Exacerbation Risk (CAPTURE) questionnaire was developed by Martinez in 2016 [13], and comprises five items with a total score of 6 points available. Scoring 0–1 represents a low risk without need for further investigations, 2–4 a moderate risk with patients recommended to follow up with a peak expiratory flow measurement with an inexpensive device, and those scoring 5–6 points advised to undergo further evaluation including spirometry [13]. It aims to identify patients with severe, high-risk, undiagnosed COPD in primary care settings.

The COPD Screening Questionnaire (COPD-SQ) was developed by Zhou et al., based on epidemiological data from 19,800 subjects in the Chinese population aged ≥40 [14]. The associated AUC was 0.829 and, on subsequent validation with 3231 subjects, the AUC was determined to be 0.812, again indicating a high accuracy. The questionnaire comprises of a seven-item assessment, also explicitly including the exposure to biomass fumes as a question, as well as other indicators of risk factor exposure, the frequency of cough and a patient subjective description of their level of dyspnoea. Although the USPSTF found no benefit in the utilization of patient screening questionnaires to uncover undiagnosed COPD in asymptomatic persons, thus not recommending screening in the general population, utilizing such questionnaires has uncovered COPD in a variety of settings. With prevalence rates of between 12 and 25%, it has been proposed to reassess the validity of these screening modalities and outcomes in the opportunistic detection in a targeted population [15].

### 2.3. Lung Function

Forced expiratory volume in the first second (FEV1) and forced vital capacity (FVC) were measured using a spirometry system (Masterscreen-PFT, Erich Jaeger, GmbH, Hoechberg, Germany), and FEV1/FVC was calculated. All lung function measurements were made strictly following the European Respiratory Society/American Thoracic Society standardization [16], both at baseline and after 20 min of the administration of 400 mcg of salbutamol through a metered-dose inhaler coupled to an Aerochamber spacer. 

### 2.4. Statistical Analysis

The Kolmogorov–Smirnov test was used to evaluate the normal distribution of the data. Continuous parameters with a normal distribution are reported as the mean ± standard deviation, while those without a normal distribution are reported as the median (interquartile range). Categorical values were analyzed using the chi-square test or Fisher’s exact test as appropriate, and were reported as *n* (%). Continuous variables were compared by Student’s *t*-test for independent samples for normally distributed data, or by Mann–Whitney U test for non-normally distributed data. Spearman’s correlation test was used to compare the questionnaires with each other and with the FEV1/FVC ratio. The optimal cut-off value of the screening questionnaire was defined as the one with the highest Youden index. Receiver operating characteristic curves (ROC) were drawn for each questionnaire. The sensitivity, specificity, positive predictive value (PPV), negative predictive value (NPV) and area under the curve (AUC) were calculated, respectively, when the optimal cut-off value and previously recommended one were taken. Sensitivity, specificity, positive and negative predictive value and AUC were calculated with SPSS software. By multiple binomial logistic regression analysis performed using the five questionnaires as independent parameters, we calculated the accuracy in predicting COPD. The screening accuracy of each questionnaire was evaluated by an AUC of 0.5–0.7, 0.7–0.9, 0.9–1.0 and 1.0 representing low, moderate, high and perfect accuracy, respectively. The sample size of at least 50 patients per group was calculated to obtain a 0.5 improvement in the minimum AUC (AUC = 0.631) of the questionnaires to predict COPD used in a previous study [17], assuming an alpha error of 0.05, a beta error of 0.20, and a study power of 0.80. Significance values were assumed for *p* < 0.050. All statistical analyses were performed using SPSS for Windows 23.0 (SPSS, Chicago, IL, USA).

## 3. Results

A total of 144 patients participated in the study. Among them, 138 individuals completed all five questionnaires, undertook spirometry, and were included in analysis (Figure 1). Table 1 represents the demographic characteristics of patients who undertook spirometry testing and who completed all five questionnaires. Comparing the COPD (*n* = 86) and non-COPD (*n* = 52) groups, no differences were reported in terms of age, smoking status and BMI. In both groups, males were around five-fold more numerous than females. As expected, lung function values were significantly lower in the COPD group (*p* < 0.01), and the mean questionnaire scoring was higher for all questionnaires in the COPD group. There was a statistically significant difference in the scoring between the 2 groups using the CDQ, LFQ and COPD-SQ questionnaires (Figure 2). 

The AUC values for the questionnaires with 95% confidence interval were as follows: CDQ = 0.714 (0.555–0.872), *p* = 0.002); LFQ = 0.331 (0.183–0.479), *p* = 0.043; COPD-PS = 0.652 (0.497–0.807), *p* = 0.021; COPD-SQ = 0.679 (0.520–0.837), *p* = 0.017; and CAPTURE = 0.602 (0.431–0.773), *p* = 0.320 (Table 2, Figure 3). The AUC findings suggest that the COPD-SQ, and COPD-PS questionnaires could be utilized for the screening of COPD with a low accuracy (0.5 < AUC < 0.7). The CDQ may also be used with a moderate accuracy (0.714). The AUC value of all five tests combined was 0.742 (0.598–0.887), with a *p* value of 0.004.

In this study, the optimal cut-off value was defined as the one with the highest Youden index. Table 2 presents the PPV and NPV values for each questionnaire at the previously recommended cut-off and optimal cut-off values. The optimal cut-off values were calculated to be: CAPTURE ≥ 2, CDQ ≥ 22, LFQ ≤ 10, COPD-PS ≥ 4, and COPD-SQ ≥ 17.

Spearman’s correlation test showed that all questionnaires correlated with each other and with the FEV1/FVC ratio (see Table 3), except for CAPTURE. The best correlation was obtained by COPD-SQ and CDQ.

## 4. Discussion

To date, several questionnaires have been developed in different populations. The realization that COPD is not simply a consequence of smoking, but that multiple environmental factors can contribute to its development, gives a novel opportunity to employ various new strategies to assist in the early recognition, diagnosis and potentially for the prevention of the disease development. The existence of precursor conditions, and the treatable traits model, also allude to the benefit of early intervention. With current confirmatory diagnosis taking approximately 6 years from symptom onset, it seems prescient to utilize simple, effective, and widely distributable tools for COPD screening.

At the present time, in Italy, the importance of COPD is still insufficiently recognized by the general public, nor is it diagnosed adequately. The immediate and long-term clinical, social, and economic impact of the disease is also undervalued. With the prevalence set to continue to rise, it seems prescient to utilize economically viable strategies and tools which can be employed in a time-saving manner to rapidly identify undiagnosed COPD [18]. GOLD guidelines from 2022 proposed the potential benefit of introducing screening questionnaires in the primary care setting. Among the 5 investigated questionnaires, this study suggests that the CDQ would be the best to employ in the early identification of patients who have COPD. The COPD-SQ also has the potential to discriminate between COPD and non-COPD patients in a Southern Italian population. The CDQ has a moderate screening accuracy, and the COPD-SQ has low accuracy, when the optimal cut-off scores are used. The COPD-PS also displayed low accuracy in screening for COPD; however, it did not have significantly different scoring between non-COPD and COPD groupings. The ROC curve demonstrated the CDQ to be the most accurate in the assessment of COPD, as well as having the greatest correlation in determining reduction in the FEV1/FVC ratio. The USPSTF has recommended the usage of the CDQ questionnaire, because it has a high diagnostic accuracy [19]. With a cut-off value of 19, when used in symptomatic patients, the CDQ has a sensitivity and specificity of 67.5% and 68.4%, from a recent study conducted by Zhou et al. [17] At the optimal cut-off of 22 in the Southern Italian population, the CDQ had a sensitivity of 88.09 and a specificity of 52.94, compared to 100% sensitivity and 11.76% specificity when 17 is taken as a cut-off. Of the five questionnaires evaluated, it was deemed to have the best capacity to detect COPD (Table 2). The CDQ in this study also had the greatest negative correlation to individuals FEV1/FVC of the five assessed, and scoring interpretation could be useful in assessing the severity of airway obstruction with moderate certainty (Table 3). Spyratos et al. compared the CDQ, COPD-PS and LFQ questionnaires, demonstrating an NPV of 94–96% and an AUC of 0.794 [20]. The CDQ could be employed in a variety of settings to screen for COPD with relative ease; however, it was the longest questionnaire to complete, requiring input from a medical professional to calculate pack years and BMI, and it should be noted that patient fatigue was an element affecting its completion. Notably, the CDQ questionnaire was shown to be useful as a screening tool in the Italian population; however, if compared with the previous literature, the other parameters tend to show better outcomes in larger group size. Undoubtedly, larger studies are required to unveil whether this might be a distinct feature of Southern Italian population compared to others. The COPD-SQ could be utilized with a cut-off of 17. Zhou et al. found that the optimal cut-off in the Chinese population was 17, with an AUC of 0.653 [18]. This study concurs with the low screening accuracy of the COPD-SQ. Another observation also reported sensitivity, specificity, and AUC of COPD SQ as 76.8%, 75.5%, 44.9% and 0.76, respectively, with a cut-off of 16 used [21]. The suggestion is that the COPD-SQ may be used to screen a larger Italian population in primary care with the cut-off value of 17. The COPD-PS, when assessed at its originally intended cut-off value of 5, yielded a sensitivity of 73.8% and specificity of 41.17%. In accordance with the intentions of the developers when the optimal cutoff of 4 was used, sensitivity increased to 97.61%, with the specificity dropping to 23.52%. To be useful as a screening tool for what is currently an underdiagnosed disease, it would be of benefit to use a threshold which yielded a higher sensitivity. In the Southern Italian population, the cut-off value of 4 could be utilized, and would provide a high sensitivity of 97.61%, which could be of greater benefit when utilizing the COPD-PS as an initial screening tool to identify potential airway obstruction, and indicating that further spirometry assessment is required. A meta-analysis conducted by Gu et al. demonstrated the AUC of the COPDPS to be 0.79, with a sensitivity of 74.52% and a specificity of 70.24% at a cut-off of 4, further validating the COPD-PS validity as a screening tool [22]. The five-item questionnaire also considers acute symptomatic changes, as well as the annual change in the patient’s functionality. This makes it particularly useful in assessing the impact on the quality of life, and the risk of acute exacerbations, both of which GOLD has put greater emphasis on establishing in their recent guidelines. Comparative analysis of the COPD-PS and SQ shows the two to have a “moderate” correlation ((R 0.506) *p* < 0.05), and are comparable in the Southern Italian population sampled when used to assess airflow obstruction. Tsukuya et al. also demonstrated the COPD-PS and CDQ as being comparable in their ability to distinguish patients with and without airflow obstruction [23]. Zhou et al., in a Chinese population, found that, of the five questionnaires utilized in this study, the LFQ was of the highest diagnostic accuracy. (0.719 (0.688–0.747)). They concluded that the CAPTURE and LFQ have a higher optimal cut-off than previously reported (3 and 16, respectively) and that these values provide higher diagnostic accuracy than previously reported [18]. In this study it can be stated that adjusting the cut-off to 10 for the LFQ would adjust the PPV to 86.36, while reducing the sensitivity to 54.76. The AUC of the LFQ in this study also suggests it lacks the accuracy to be used as a screening tool in the Italian population. The CAPTURE contains only five items, and is relatively quick to complete. However, in this study it cannot be considered useful, due to its inability to distinguish between COPD and non-COPD groups, and its low AUC value. Both the CAPTURE and LFQ had the lowest FEV1/FVC correlation index of −0.206 and −0.316, indicating that, of the five questionnaires, they were the weakest at predicting FEV1/FVC ratios and subsequent severity of COPD [17].

The most important limitation of the study is the limited population sample size. In future, enrollment will have to be increased in order to allow the validation of the results in the general population. The fact that all questionnaires were completed simultaneously invariably leads to fatigue on the part of the patient and inaccurate reporting. It was also noted that a small percentage of patients struggled to complete the questionnaires due to illiteracy, and required prompting to allow for questionnaire completion. Invariably, this increases the risk of introducing operator bias into the study, and should be noted if considering the usage of questionnaires to accurately follow diagnosed COPD patients’ responses to treatment interventions. Particular comments referred to the challenging wording of certain questions and the requirement of patients to calculate BMI or smoking status pack years. Another is that the benefit of completing questionnaires is not always perceived by the patient, unlike the undertaking of more formal testing, and this may influence their attitude and subsequent response to the interventions offered. Future studies should attempt to integrate the usage of simple physiological assessment tools, e.g., peak flow assessment with screening questionnaires, in order to assess for improved sensitivity of COPD diagnosis. Longer-term management follow up, particularly once treatable traits have been identified in the patient group, should be evaluated, and it would be prudent to assess the adaptability of the screening questionnaires in the future to be utilized as a reinforcing educational, behavioral, and monitoring tool relating to treatable trait modification. This approach may be especially prudent, considering that both a Cochrane review and subsequent meta-analysis concluded that hospital readmission rate due to moderate to severe COPD exacerbations can be reduced, and better self-management achieved in mild COPD patients [24]. 

In the general population, the role of spirometry for the screening of COPD is controversial. Supporting data is weak, in reference to improving COPD outcomes or in guiding management decision-making. GOLD, therefore, does not recommend spirometry in asymptomatic individuals who have no “significant” exposure risk factors, as opposed to those with symptoms and/or risk factors (>20 year pack history, recurrent chest infections, impaired lung development) for whom the diagnostic yield following spirometry assessment is relatively high. If COPD is diagnosed at an early stage, and risk factors are modified or eliminated completely, then the predicted functional pulmonary decline can be decreased dramatically [25]. Novel approaches to screening, therefore, are being considered and adapted, which attempt to embrace exposure risk, patients’ symptoms and combine with simple peak flow measurements. In various settings, these approaches can identify previously undiagnosed COPD, usually in patients who have mild to moderate disease.

## 5. Conclusions

To conclude, of the five questionnaires assessed, the CDQ questionnaire is useful as a screening tool in the Southern Italian population. It is an eight-item questionnaire, the longest of those assessed and, as an alternative, less accurate screening tool for COPD, the COPD-SQ could be utilized to assist in identifying patients with undiagnosed COPD.

## Figures and Tables

**Figure 1 medicina-59-01252-f001:**
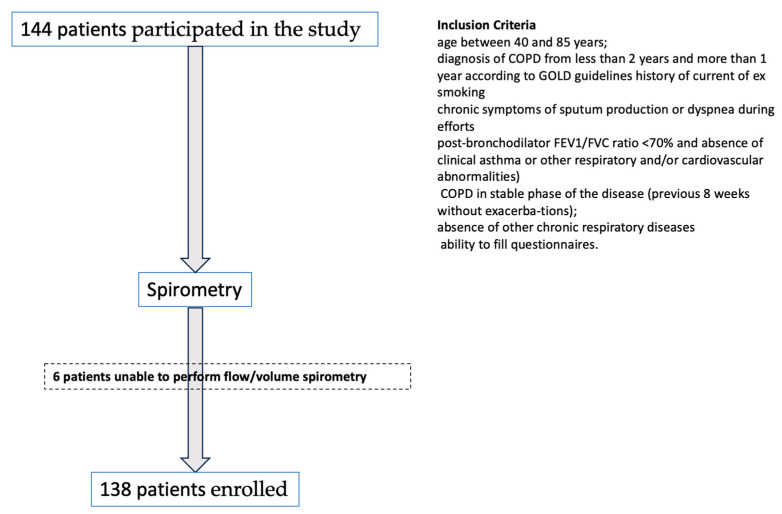
Flow chart of the study.

**Figure 2 medicina-59-01252-f002:**
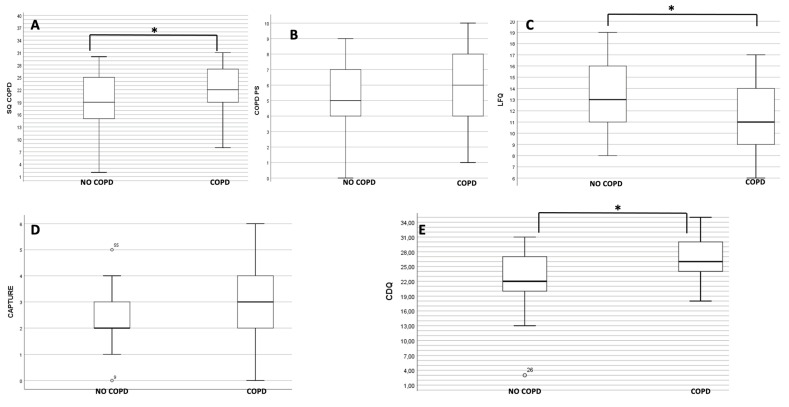
Box plot graphs for each screening test. (**A**) Comparison between COPD and non-COPD population for the SQ-COPD screening test. (**B**) Comparison between COPD and non-COPD population for the PS-COPD screening test. (**C**) Comparison between COPD and non-COPD population for the LFQ screening test. (**D**) Comparison between COPD and non-COPD population for the CAPTURE screening test. (**E**) Comparison between COPD and non-COPD population for the CDQ screening test. *: *p*-value < 0.05. Abbreviations: CDQ: COPD Diagnostic Questionnaire; COPD-PS: COPD Population Screener; COPD-SQ: COPD Screening Questionnaire; CAPTURE: COPD Assessment in Primary Care to Identify Undiagnosed Respiratory Disease and Exacerbation Risk; LFQ: the Lung Function Questionnaire.

**Figure 3 medicina-59-01252-f003:**
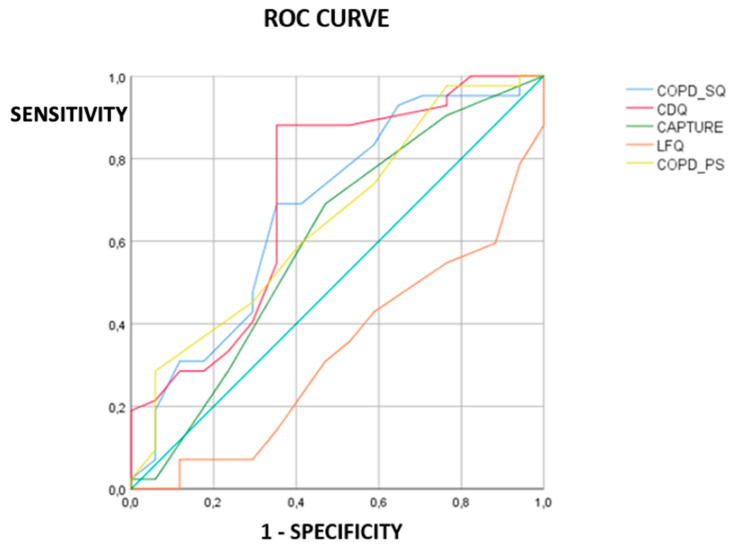
ROC curves for the five questionnaires using the study population. Abbreviations: CDQ: COPD Diagnostic Questionnaire; COPD-PS: COPD Population Screener; COPD-SQ: COPD Screening Questionnaire; CAPTURE: COPD assessment in Primary Care to Identify Undiagnosed Respiratory Disease and Exacerbation Risk. LFQ: the Lung function questionnaire.

**Table 1 medicina-59-01252-t001:** Comparison between COPD and non-COPD population.

	COPD (*n* = 86)	Non-COPD (*n* = 52)	*p*-Value
Age (years) m ± sd	70.74 ± 7.16	67.17 ± 12.24	ns
Sex (M/F)	67/19	42/10	ns
BMI (kg/m^2^) m ± sd	28.42 ± 5.88	29.42 ± 3.87	ns
Pack-year	44.8 ± 21.9	42.3 ± 16.5	ns
Current smokers %	55	62	ns
FEV1 post-BD (L)	1.62 ± 0.59	3.20 ± 0.66	<0.01
FEV1/FVC post-BD	0.47 ± 0.12	0.78 ± 0.08	<0.01
CAPTURE	3.0 (2.0–4.0)	2.0 (1.5–3.5)	ns
CDQ	26.0 (24.0–30.0)	22.0 (19.0–27.5)	<0.05
LFQ	10.0 (8.0–12.0)	14.0 (12.0–17.0)	<0.05
COPD-PS	6.0 (4.0–8.0)	5.0 (3.5–7.5)	ns
COPD-SQ	22.0 (19.5–27.5)	19.0 (15.0–25.0)	<0.05

For age, BMI, FEV1 and FEV1/FVC values are intended as mean ± standard deviation. BMI = body mass index. ns = not significant.

**Table 2 medicina-59-01252-t002:** Characteristics of the five COPD screening questionnaires.

Questionnaires	AUC	CutOff	Sensitivity	Specificity	PPV	NPV	*p*-Value
CAPTURE	0.602 (0.431–0.773)	≥2	90.47	17.64	73.07	42.85	ns
		≥2 *	90.47	17.64	73.07	42.85	ns
CDQ	0.714 (0.555–0.872)	≥17	100	11.76	73.68	100	ns
		≥22 *	88.09	52.94	82.22	64.28	<0.01
LFQ	0.669 (0.521–0.817)	≤18	100	5.88	72.4	100	ns
		≤10 *	54.76	82.35	86.36	37.88	<0.05
COPD-PS	0.652 (0.497–0.807)	≥5	73.80	41.17	75.60	38.8	ns
		≥4 *	97.61	23.52	75.92	80.0	<0.05
COPD-SQ	0.679 (0.520–0.837)	≥17	95.23	29.41	76.92	71.42	<0.05
		≥21 *	66.66	58.82	80.0	41.66	ns

*: new cut-off (Youden index value). Abbreviations: CDQ: COPD Diagnostic Questionnaire; COPD-PS: COPD Population Screener; COPD-SQ: COPD Screening Questionnaire; CAPTURE: COPD Assessment in Primary Care to Identify Undiagnosed Respiratory Disease and Exacerbation Risk; LFQ: the Lung Function Questionnaire; PPV, positive predictive value; NPV, negative predictive value; AUC, the area under the curve. ns = not significant.

**Table 3 medicina-59-01252-t003:** Spearman correlation between questionnaires and FEV1/FVC ratio.

	CAPTURE R (*p*-Value)	CDQ R (*p*-Value)	LFQ R (*p*-Value)	COPD PS R (*p*-Value)	COPD SQ R (*p*-Value)	FEV1/FVC R (*p*-Value)
CAPTURE		0.128 (0.333)	−0.364 (0.005)	0.473 (0.000)	0.403 (0.002)	−0.206 (0.118)
CDQ	0.128 (0.333)		−0.456 (0.000)	0.303 (0.020)	0.622 (0.000)	−0.426 (0.001)
LFQ	−0.364 (0.005)	−0.456 (0.000)		−0.638 (0.000)	−0.713 (0.000)	−0.316 (0.015)
COPD-PS	0.473 (0.000)	0.303 (0.020)	−0.638 (0.000)		0.506 (0.000)	−0.334(0.015)
COPD-SQ	0.403 (0.002)	0.622 (0.000)	−0.713 (0.000)	0.506 (0.000)		−0.381 (0.003)
FEV1/FVC	−0.206 (0.118)	−0.426 (0.001)	0.316 (0.015)	−0.334(0.015)	−0.381 (0.003)	

Abbreviations: R of Spearman (*p*-value). CDQ: COPD Diagnostic Questionnaire; COPD-PS: COPD Population Screener; COPD-SQ: COPD Screening Questionnaire; CAPTURE: COPD screening questionnaire; LFQ; Lung function questionnaire.

## Data Availability

Anonymized dataset can be provided at request.

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
