# Peer review of "Assessment of Five Questionnaires for Chronic Obstructive Pulmonary Disease in a Southern Italian Population: A Proof-of-Concept Study"

_medicina, 2023, doi:10.3390/medicina59071252_

Round 1

Reviewer 1 Report

The present findings suggest that questionnaire-based screening for COPD may help in the early diagnosis of this disease in respect to Italian population. However, the study was conducted extensively but the greatest limitation of the study is population size. There are certain recommendations mentioned below:

1.      The authors must provide new and valid citations to show the global incidence or prevalence in the introduction section. They may cite organizations like WHO or newer data from such agencies. Do the authors refer to world data in line 35-36? Again, in line 37-38, a relatively very old data of 2002 has been incorporated to show the Italian aspect. Kindly cite new data. The line 29-31, cite a valid citation.

2.      What are the possible reasons for underdiagnosis and undertreatment of COPD in Italy as the global stats suggest that life expectancy is relatively higher. Mention it in the text.

3.      A strong rationale is lacking in the ending part of introduction explaining why this study was important to conduct. It must support the whole idea reasonably.

4.      Add to introduction, what are various factors, tests available or diagnosis of COPD? Spirometry yet controversial but widely used to diagnose COPD and related disorders. Comment?

5.      Mention in 2.1 about the age selection for the study and male to female differences. The range of age should also be mentioned in abstract. This is an important aspect which is greatly missed.

6.      What were the parameters chosen for sample size? For such studies, the patient participants (138) are extremely low so as to cover a larger group. Larger studies provide stronger and more reliable results because they have smaller margins of error and lower standards of deviation.

7.      If possible, add a flow chat for the study design as it would be more reader friendly.

8.      The table 1, shows the comparison between COPD and non-COPD population. Why was the smoking status not included in this table?

9.      The sensitivity in Figure 1, should be written in correct manner as per the axis.

10.  Youden index values were not shown or discussed in the manuscript. Why?

11.  A very recent study by Zhou et al, 2022 (doi:10.2147/COPD.S341648) suggest the Accuracy of Six COPD Screening Questionnaires in the Chinese Population. Moreover, the sample size is relatively very high. So, how different is this current study from the mentioned? Demography? It seems like the current work is replica of mentioned work with a change in demography.

12.  CDQ questionnaire was shown useful as a screening tool in the Italian population however, if compared with the previous literature, the other parameters tend to show better outcomes in larger group size. How can the authors justify the same using smaller population?

Grammar correction is must. The authors must rectify the English grammar errors and typos.

Author Response

The present findings suggest that questionnaire-based screening for COPD may help in the early diagnosis of this disease in respect to Italian population. However, the study was conducted extensively but the greatest limitation of the study is population size. There are certain recommendations mentioned below:

  1. The authors must provide new and valid citations to show the global incidence or prevalence in the introduction section. They may cite organizations like WHO or newer data from such agencies. Do the authors refer to world data in line 35-36? Again, in line 37-38, a relatively very old data of 2002 has been incorporated to show the Italian aspect. Kindly cite new data. The line 29-31, cite a valid citation.

R- We agree with the reviewer about updating references with more recent ones. We have now modified references 1-4 and 5 (please see the manuscript).

  1. What are the possible reasons for underdiagnosis and undertreatment of COPD in Italy as the global stats suggest that life expectancy is relatively higher. Mention it in the text.

R- Thanks for your comment. We have now added a detailed explanation of the reasons for underdiagnosis and undertreatment of COPD in Italy.

  1. A strong rationale is lacking in the ending part of introduction explaining why this study was important to conduct. It must support the whole idea reasonably.

R- Thanks for your comment. We have now added a strong rationale for the implementation of screening tests in COPD.

  1. Add to introduction, what are various factors, tests available or diagnosis of COPD? Spirometry yet controversial but widely used to diagnose COPD and related disorders. Comment?

R- Thanks for your comment. We have now implemented introduction section according to your suggestions.

  1. Mention in 2.1 about the age selection for the study and male to female differences. The range of age should also be mentioned in abstract. This is an important aspect which is greatly missed.

R- We apologize with the reviewer for these missing data. We have now added age range in the text, as well as gender data in Table 1 and results.

  1. What were the parameters chosen for sample size? For such studies, the patient participants (138) are extremely low so as to cover a larger group. Larger studies provide stronger and more reliable results because they have smaller margins of error and lower standards of deviation.

R-Thanks for your comment. We have now implemented statistical analysis section according to your suggestions. The sample size of at least 50 patients per group was calculated to obtain a 0.5 improvement in the minimum AUC (AUC=0.631) of the questionnaires to predict COPD used in a previous study [18], assuming an alpha error of 0 .05, a beta error of 0.20, and a study power of 0.80.

  1. If possible, add a flow chat for the study design as it would be more reader friendly.

R- Thanks for the comments. The study design flow chart has been added.      

  1. The table 1, shows the comparison between COPD and non-COPD population. Why was the smoking status not included in this table?

R- We apologize with the reviewer for missing data. We have now included smoking status data in table 1 and in results section.

  1. The sensitivity in Figure 1, should be written in correct manner as per the axis.

R- Thanks for the comment. We have made the recommended change.

  1. Youden index values were not shown or discussed in the manuscript. Why?

R- Thanks to the reviewer for the comment. The new cutoffs were obtained with the Youden index. We proceeded to specify it in table number 2.

  1. A very recent study by Zhou et al, 2022 (doi:10.2147/COPD.S341648) suggest the Accuracy of Six COPD Screening Questionnaires in the Chinese Population. Moreover, the sample size is relatively very high. So, how different is this current study from the mentioned? Demography? It seems like the current work is replica of mentioned work with a change in demography.

R- Thanks for your interesting comment. The study has a very similar design to that of Zhou et al. However, the population of patients is totally different, since the important differences of ethnicity have to be taken into account. Indeed, the study of Zhou, which has been already cited in our manuscript, includes one more questionnaire, which was specifically designed for Chinese population, and which would not be applicable in an European group of patients. Undoubtedly, the main limitation of our study is the limited sample size and larger studies are mandatory to confirm our data. 

  1. CDQ questionnaire was shown useful as a screening tool in the Italian population however, if compared with the previous literature, the other parameters tend to show better outcomes in larger group size. How can the authors justify the same using smaller population?

R- This is a very interesting comment. Many thanks for it. We also were wondered about our findings. To date, we cannot state whether it is related to a specific feature of Italian population and larger studies are required. We have now addressed this point in discussion section.

Comments on the Quality of English Language

Grammar correction is must. The authors must rectify the English grammar errors and typos.

R- Many thanks for noticing it. We have performed a detailed grammar and typos correction.

Reviewer 2 Report

This manuscript by Silvano et al. summarized five questionnaires from publications, including CDQ, LFQ. COPD-PS, COPD-SQ and CAPTURE for COPD identification and assessed mainly about their sensitivity and specificity in identifying patients with potential COPD in Italian population. Results showed CDQ and COPD-SQ questionnaires work the best to predict the patients with COPD, which was confirmed by FEV1/FVC ratio. Besides, the cut-off values for each questionnaire were also optimized. This research is meaningful, especially for lung disease surveillance and prevention in impoverished areas. The written of this manuscript needs improvement, so as data presentation.

Specific comments:

1.     Table 1: Table 1 should display the scores from 138 participants after each questionnaire in a separate column figure, with dots representing individual values. This way, the score range of each method can be easily observed, and the main cluster area of the score for each group can be displayed directly.

2.     Table 1: The p-value of COPD PS is shown as "ns" in Table 1, while it is quoted as "p=0.021" in line 182. It needs to be confirmed which one is correct.

3.     Table 1: The "COPD score (range)" in both CDQ and LFQ is the same, as shown as "26.0 (24.0-30.0)". Are these numbers correct? If yes, regarding the choice of the cutoff value, the score range of COPD and Non-COPD patients in LFQ is distinct, with scores greater than 24 in COPD patients and less than 16 in Non-COPD patients, as shown in Table 1. This suggests that choosing a cutoff value between 16 and 24 may provide better specificity, as well as positive predictive value (PPV) and negative predictive value (NPV). However, it is unclear why the optimized cutoff value was determined as “10”, as shown in Table 2. Please provide an explanation or clarification for this optimization in your manuscript.

4.     Figure 1: The method used for calculating the values of "Specificity" and "Sensitivity" for each questionnaire should be clearly mentioned in the Method. Please provide the specific software used for calculating these values in your manuscript if used.

5.     Add explanation about  the meaning of "+" and "-" in the Spearman correlation values shown in Table 3, as "+" and "-" symbols indicate the direction of the correlation as well as the correlation values themselves indicate the strength of the relationship, ranging from -1 to +1.

Minor comments:

1.     To maintain consistency in the manuscript, it is important to use consistent terminology and abbreviations. In this case, “COPD-PS” (line 12) & “COPD PS”( line19)& “COPD_PS”( Figure 1) , “COPDPS”(line 271) should be used consistently throughout the manuscript. Similarly, for other abbreviations like "COPD-SQ" and "COPD SQ," the same format should be used consistently.

2.     Should clarify the age range of your patients participated in this research is >40y in your “Patients and study design”.

3.     Line 183:  Should correct the quote from “(Figure 1)” to “(Table 2)”;

4.     Table 2: There is a discrepancy in the presentation of the cutoff values between the LFQ group and the other four methods. The LFQ group shows the cutoff value as "less than," while the other methods show it as "more than." This inconsistency may be confusing and needs to be addressed.

5.     Table 3: Should simplify the table by removing the repeated data shown on both sides of the blank space. Table 3: The correct COPD SQ correlation value with LFQ should be "-0.713 (0.000)" instead of "-0713 (0.000)". Table 3 and line 291: There is a discrepancy in the correlation value of LFQ with FEV1/FVC ratio. In Table 3, it is stated as "0.316 (0.015)," while in line 291 it is mentioned as "-0.316." It is important to verify which value is correct and ensure consistency between the Table and the text.

6.     Line 248: Quote “(Table 2)” after “……have the best capacity to detect COPD”;

7.     Line 251: Quote “(Table 3)” after “…… with moderate certainty.”;

8.     Line 259: Followed “……sensitivity, specificity, and AUC of COPD SQ……” showed 4 numbers. Should delete an extra number.

Author Response

This manuscript by Silvano et al. summarized five questionnaires from publications, including CDQ, LFQ. COPD-PS, COPD-SQ and CAPTURE for COPD identification and assessed mainly about their sensitivity and specificity in identifying patients with potential COPD in Italian population. Results showed CDQ and COPD-SQ questionnaires work the best to predict the patients with COPD, which was confirmed by FEV1/FVC ratio. Besides, the cut-off values for each questionnaire were also optimized. This research is meaningful, especially for lung disease surveillance and prevention in impoverished areas. The written of this manuscript needs improvement, so as data presentation. 

Specific comments:

  1. Table 1: Table 1 should display the scores from 138 participants after each questionnaire in a separate column figure, with dots representing individual values. This way, the score range of each method can be easily observed, and the main cluster area of the score for each group can be displayed directly. 

R- Thanks for your suggestion. We have now included a box plot graph for each screening test.

  1. Table 1: The p-value of COPD PS is shown as "ns" in Table 1, while it is quoted as "p=0.021" in line 182. It needs to be confirmed which one is correct.

R- Thanks for the comment. P-value in table 1 is ns and it’s about median scores, whereas the p=0.021 refers to AUC.  

  1. Table 1: The "COPD score (range)" in both CDQ and LFQ is the same, as shown as "26.0 (24.0-30.0)". Are these numbers correct? If yes, regarding the choice of the cutoff value, the score range of COPD and Non-COPD patients in LFQ is distinct, with scores greater than 24 in COPD patients and less than 16 in Non-COPD patients, as shown in Table 1. This suggests that choosing a cutoff value between 16 and 24 may provide better specificity, as well as positive predictive value (PPV) and negative predictive value (NPV). However, it is unclear why the optimized cutoff value was determined as “10”, as shown in Table 2. Please provide an explanation or clarification for this optimization in your manuscript.

R- Thanks for the comment. We apologize for the transcription error. We proceeded to modify the table by inserting the correct value for the LFQ questionnaire.

  1. Figure 1: The method used for calculating the values of "Specificity" and "Sensitivity" for each questionnaire should be clearly mentioned in the Method. Please provide the specific software used for calculating these values in your manuscript if used.

R- Thanks for the comment. Sensitivity, specificity, positive and negative predictive value were calculated with Spss software version 23. We specified it in the statistical analysis section.

  1. Add explanation about  the meaning of "+" and "-" in the Spearman correlation values shown in Table 3, as "+" and "-" symbols indicate the direction of the correlation as well as the correlation values themselves indicate the strength of the relationship, ranging from -1 to +1.

R- Thanks for the comment that allows us to make an important clarification. The "+" indicates the presence of a directly proportional correlation whereas the "-" indicates the presence of an inversely proportional correlation. We have updated the table.

Minor comments:

  1. To maintain consistency in the manuscript, it is important to use consistent terminology and abbreviations. In this case, “COPD-PS” (line 12) & “COPD PS”( line19)& “COPD_PS”( Figure 1) , “COPDPS”(line 271) should be used consistently throughout the manuscript. Similarly, for other abbreviations like "COPD-SQ" and "COPD SQ," the same format should be used consistently. 

R- Thanks for your comment. We have harmonized the terminology throughout the manuscript.

  1. Should clarify the age range of your patients participated in this research is >40y in your “Patients and study design”.

R- We apologize for the missing data. Age range is now in table 1 and methods section.

  1. Line 183:  Should correct the quote from “(Figure 1)” to “(Table 2)”;

R- Thanks for noticing the inaccuracy. We have modified it.

  1. Table 2: There is a discrepancy in the presentation of the cutoff values between the LFQ group and the other four methods. The LFQ group shows the cutoff value as "less than," while the other methods show it as "more than." This inconsistency may be confusing and needs to be addressed.

R- Thanks for this very interesting comment. Indeed, this is not an inconsistency since LFQ is the only test among the five ones which has been designed to have cutoff values with “less than” instead of “more than”. Therefore, we decided to preserve its integrity when attempting to find new cutoff values for our population. For avoid confusion to readers, we have now addressed this issue in methods section.

  1. Table 3: Should simplify the table by removing the repeated data shown on both sides of the blank space. Table 3: The correct COPD SQ correlation value with LFQ should be "-0.713 (0.000)" instead of "-0713 (0.000)". Table 3 and line 291: There is a discrepancy in the correlation value of LFQ with FEV1/FVC ratio. In Table 3, it is stated as "0.316 (0.015)," while in line 291 it is mentioned as "-0.316." It is important to verify which value is correct and ensure consistency between the Table and the text.

R- Thanks for your very detailed check which allowed us to detect several errors in table 3 which have now been corrected.

  1. Line 248: Quote “(Table 2)” after “……have the best capacity to detect COPD”;

        R-Done it. Thank you.

  1. Line 251: Quote “(Table 3)” after “…… with moderate certainty.”;

        R-Done it. Thank you.

  1. Line 259: Followed “……sensitivity, specificity, and AUC of COPD SQ……” showed 4 numbers. Should delete an extra number.

        R-Done it. Thank you.

Reviewer 3 Report

1.The inclusion criteria should be described in the Methods section.

2.The sample size is relatively small and how do you decide the sample size?

3.The diagnostic criteria of COPD should be described.

4. There are some format mistakes in the Tables, Figures and data presentation that should be corrected.

5. AUC <0.5 is rarely seen and LFQ for COPD in this study is 0.331. Please further address the reasons and explanation for this result.

6. Whether combination of 5 questionnaires may derive a better diagnostic value is suggested to be evaluated.

7. Please introduce the representative of the study population. Whether it’s sufficient to represent the whole Italian population.

8. The discussion section is suggested to be more logically.

The manuscript is generally easy to understand and the logic manner could be further improved.

Author Response

1.The inclusion criteria should be described in the Methods section.

R- Thanks for your comment. We have now added inclusion criteria in methods section.

2.The sample size is relatively small and how do you decide the sample size?

R- Thanks for your comment. We have now implemented statistical analysis section according to your suggestions. The sample size of at least 50 patients per group was calculated to obtain a 0.5 improvement in the minimum AUC (AUC=0.631) of the questionnaires to predict COPD used in a previous study [18], assuming an alpha error of 0 .05, a beta error of 0.20, and a study power of 0.80.

3.The diagnostic criteria of COPD should be described.

R- Thanks for your comment. We have now included diagnostic criteria of COPD in methods section.

  1. There are some format mistakes in the Tables, Figures and data presentation that should be corrected.

R- Many thanks for noticing it. We have corrected several inaccuracies in all figures and tables.

  1. AUC <0.5 is rarely seen and LFQ for COPD in this study is 0.331. Please further address the reasons and explanation for this result.

R- Thanks to the reviewer for the comment. Unfortunately,  an error was made while calculating the LFQ AUC by reversing the "polarity" of the calculation. We have updated the table with the correct value. We apologize for that.

  1. Whether combination of 5 questionnaires may derive a better diagnostic value is suggested to be evaluated.

R- We thank the reviewer for the interesting and welcome comment which allows us to add to the manuscript a result that we believe is interesting. By multiple binomial logistic regression analysis performed using the 5 questionnaires as independent parameters, we calculated the accuracy in predicting COPD.The AUC value of all 5 tests combined was 0.742 [0.598 - 0.887] with a p value of 0.004.

We have updated the statistical analysis and results section.

  1. Please introduce the representative of the study population. Whether it’s sufficient to represent the whole Italian population.

R- Thanks for your very interesting comment. Our outpatient clinic gathers is one of the largest in south Italy and gathers patients coming from all southern Italy regions. Therefore, we believe that sample size could be sufficient to represent the southern italian population and not the whole Italian people. Thus, we have modified the title and the text accordingly.

  1. The discussion section is suggested to be more logically.

R- Thanks for your suggestion. We have modified the discussion and conclusion session in a more logical way.

Round 2

Reviewer 1 Report

The manuscript entitled, "Assessment of five questionnaires for Chronic Obstructive Pulmonary Disease in a Southern Italian population" has been thoroughly revised by the authors and all the comments are taken care appreciably. The work showed an extensive study involving the questionnaire for early screening of COPD  and thus will help for its diagnosis. The only suggestion is to increase population size/sample number to generally validate the problem in a broader manner. This can be one of the future direction and really necessary one.  The present modifications make it suitable to be accepted for publication.

Author Response

We thank reviewer 1 for the comments. We agree with him/her on the importance of the small sample size. In the future, enrollment will have to be increased to allow validation of the results in the general population. For this reason, we have changed the title by introducing the term “A Proof-of-concept study”. At line number 367 in the discussion section, we have introduced the following sentence: The most important limitation of the study is the limited population sample size. In the future, enrollment will have to be increased in order to allow validation of the results in the general population

Reviewer 2 Report

1. Figure 2: Should mark each panel as “(A), (B)…(E)”, and add figure legend to briefly introduce the figure and explain what “*” represent for. In the first panel (LFQ), also need to add the p value between COPD& NO COPD. Besides, the five panels of Figure 2 need to rearranged, especially panel COPD PS. 

Author Response

Thanks for the comment. We have updated table number 2.

Reviewer 3 Report

The authors have addressed my comments satisfactorily.

Author Response

Thanks for the welcome comment